# Porphyrins and Metalloporphyrins Combined with N-Heterocyclic Carbene (NHC) Gold(I) Complexes for Photodynamic Therapy Application: What Is the Weight of the Heavy Atom Effect?

**DOI:** 10.3390/molecules27134046

**Published:** 2022-06-23

**Authors:** Stefano Scoditti, Francesco Chiodo, Gloria Mazzone, Sébastien Richeter, Emilia Sicilia

**Affiliations:** 1Department of Chemistry and Chemical Technologies, Università della Calabria, 87036 Arcavacata di Rende, Italy; stefano.scoditti@unical.it (S.S.); francescochiodo22@gmail.com (F.C.); 2ICGM, Univ. Montpellier, CNRS, ENSCM, 34293 Montpellier, France; sebastien.richeter@umontpellier.fr

**Keywords:** PDT, porphyrins, metalloporphyrins, gold complexes, carbenes, DFT, TDDFT

## Abstract

The photophysical properties of two classes of porphyrins and metalloporphyrins linked to N-heterocyclic carbene (NHC) Au(I) complexes have been investigated by means of density functional theory and its time-dependent extension for their potential application in photodynamic therapy. For this purpose, the absorption spectra, the singlet–triplet energy gaps, and the spin–orbit coupling (SOC) constants have been determined. The obtained results show that all the studied compounds possess the appropriate properties to generate cytotoxic singlet molecular oxygen, and consequently, they can be employed as photosensitizers in photodynamic therapy. Nevertheless, on the basis of the computed SOCs and the analysis of the metal contribution to the involved molecular orbitals, a different influence in terms of the heavy atom effect in promoting the intersystem crossing process has been found as a function of the identity of the metal center and its position in the center of the porphyrin core or linked to the peripheral NHC.

## 1. Introduction

Photodynamic therapy (PDT) is a clinical therapeutic modality with considerable potential for application in a wide range of diseases, such as cancer [1] as well as the antimicrobial and environmental fields [2,3,4,5,6,7]. A PDT protocol involves the combination of three key components: light, with an appropriate wavelength, molecular oxygen, and a photosensitizer, a molecule which can produce reactive oxygen species (ROS) that are deadly to cells when excited by light of a proper wavelength in presence of oxygen. The production of ROS may occur through two pathways, named type I and type II processes. The type I mechanism involves an electron-transfer-mediated biomolecule oxidation that generates reactive oxygen species such as O_2_^•−^ and HO_2_^•^. The type II mechanism entails an energy transfer from the photosensitizer in its triplet state to molecular oxygen, promoting the formation of singlet oxygen, ^1^O_2_. An efficient photosensitizer must possess a series of required chemical and photophysical properties, and some of them are of paramount importance: (i) a maximum absorption wavelength falling in the photodynamic therapeutic window (600–850 nm) to ensure a good penetration of light into living tissues [8,9]; (ii) a high intersystem spin crossing (ISC) probability between the excited singlet and triplet states; and (iii) a populated triplet state with an energy higher than 0.98 eV, the energy necessary to induce singlet oxygen formation from the ground triplet state [10,11,12,13].

Since the discovery of the potential of light-based therapies, numerous attempts have been made to treat tumors with photosensitizing agents. Porphyrins and their derivatives, such as chlorins or bacteriochlorins, surely represent one of the best classes of molecules for PDT application due to their inherent chemical stability and suitable photophysical properties [14,15,16,17,18,19]. They show characteristic UV-vis spectra composed of one strong absorption band, named the Soret or B band, in the ~380–420 nm region and weaker absorption bands, known as Q bands [20,21]. The number and the intensity of these Q bands strongly depend on the structure of the porphyrin core and the nature of the metal ion in the macrocycle. Free-base porphyrins generally possess four Q bands with the lowest energy one centered at ~650 nm. Although the molar absorption coefficients of Q bands are lower compared to the Soret band, the excitation of the porphyrin derivatives in these Q-bands is extensively used for PDT since they fall within the therapeutic window, enabling a deep penetration of red light into living tissues. Despite the excellent properties, such as their low dark toxicity, their thermodynamic stability, and their maximum absorption bands in the red region of the visible spectrum, the application of porphyrin derivatives is limited by shortcomings such as low water solubility and poor accumulation into cancer cells [22,23,24,25,26]. Over the years, several structural changes have been proposed to improve both their PDT efficiency and water solubility, which include the modification of the π electron architecture, the addition of charged groups, and the inclusion of heavy atoms [27]. It is well-known, indeed, that the presence of heavy atoms in the molecular structure of the chromophore significantly increases the spin–orbit coupling (SOC), a relativistic effect that causes a quantum mechanical mixing between states with different spin multiplicities. This phenomenon is known as the “Heavy Atom Effect” (HAE). Therefore, as established by the Fermi golden rule [28,29], high SOC values improve the kinetics of the transitions between these states, favoring the population of a triplet state required for PDT action. In order to increase the quantum yield of the triplet state, two different strategies have been proposed for porphyrinoid dyes: (i) including a metal center within the cavity of the tetrapyrrole macrocycle and (ii) linking a peripheral metal complex to the π system of the chromophore. Many studies, both experimental and computational, have been carried out dealing with such an effect for numerous metalloporphyrins [13,18,30,31,32]. For PDT applications, the peripheral metal complex might increase the SOC values and consequently boost the kinetics of the ISC, enhancing the generation of ^1^O_2_ through the HAE. For example, Spingler, Gasser, and coworkers also reported that ^1^O_2_ quantum yields of *meso* tetra(pyridyl)porphyrins are increased upon the coordination of Pt(II) complexes, suggesting that the Pt(II) ions promote the formation of the T1 state through the HAE [33]. The peripheral metal complexes can also bring additional properties such as intrinsic anticancerous properties. Thus, Brunner and coworkers reported the synthesis of porphyrin derivatives with peripheral Pt(II) complexes for dual chemo- and phototherapy because they combine within the same molecular species the antitumor activity of Pt(II) complexes and the phototoxicity of porphyrins upon irradiation with light [34,35,36,37,38].

N-heterocyclic carbenes (NHCs) are important ligands in the field of organometallic chemistry, and several molecular systems combining porphyrins and NHC-metal complexes were investigated during the last decade for different applications, including catalysis and PDT [39]. In this framework, we reported the synthesis of porphyrins with peripheral NHC-Au(I) complexes and their use as photosensitizers for PDT [39,40,41]. Photophysical measurements showed that peripheral Au(I) complexes improve PDT efficiency [40]. The obtained Au(I) also presents moderate cytotoxicity in the dark [39,40]. Interestingly, we and others also showed that the anticancerous properties of NHC-Au(I) complexes can be significantly improved by changing the second ancillary ligand of the linear Au(I) complex [42,43]. To have a better comprehension of the photophysical properties of porphyrins equipped with peripheral NHC-Au(I) complexes and their photodynamic properties, we report the computational evaluation of the photophysical properties of two sets of porphyrins, starting from the corresponding free-base porphyrins (Fb) named **Fused-Fb** and **Meso-Fb**. In the former case, the porphyrin and the NHCs are fused together and two neighboring *β*-pyrrolic carbon atoms of the porphyrin core correspond to the C4 and C5 atoms of the imidazole-2-ylidene. In the second case, the porphyrin is a wingtip group of the NHC, and both are linked together through the formation of a single C*_meso_*−N_NHC_ bond. The structures of these ligands are depicted in Figure 1 together with the adopted nomenclature for the **Fused** and **Meso** species in panels a and b, respectively. The corresponding Au(I) complexes have also been studied to establish how much the presence of the Au(I) ion can enhance the ISC conversion. Additionally, we propose for both systems and their Au(I) complexes the corresponding metalloporphyrins with Zn(II) and Pd(II) as inner metal ions (see Figure 1) based on the hypothesis that the simultaneous presence of inner and outer metal ions such as Zn(II)/Au(I) or Pd(II)/Au(I) couples could synergistically increase the SOC values and the rate of the ISC process. Density functional theory and its time-dependent formulation, TDDFT, have been employed in order to compute the photophysical properties of the two series of porphyrins and compare them with the available experimental counterparts.

## 2. Results and Discussion

Optimized geometrical structures of the two compounds named **Fused-FbAu** and **Meso-FbAu** are reported in Figure 1 along with some key geometrical parameters. To allow an easier visualization and highlight the distortion of the macrocycle caused by the fused NHC ligand of the **Fused** species, only the porphyrin cores and the NHC-Au(I) complexes are reported. The addition in the *meso* position of the NHC moiety, which is perpendicular to the porphyrin plane, does not provoke any geometrical change worth mentioning. The optimized structure of the free porphyrin fused with the NHC group has been superimposed with the experimental counterpart, as reported in Appendix A, which shows a very good agreement. Notwithstanding the observed deviation from the planarity of the **Fused** porphyrin compounds, the π conjugation in the central core is not disturbed, and the main characteristics of the spectra of the unperturbed porphyrins are conserved, as will be illustrated in the next paragraphs.

The rationalization of the UV-vis absorption spectra of porphyrin-like compounds can be conducted in terms of the classical four-orbital model first applied by Gouterman [44]. On the basis of such model, indeed, both the strong Soret band (B) in the near-UV region and the low-intensity Q-bands in the visible or near-IR region are mainly generated by the mixed electronic transitions between the two highest occupied molecular orbitals (HOMO and HOMO-1) and the two lowest unoccupied ones (LUMO and LUMO+1), here named H, H-1, L, and L+1, respectively. It has been underscored above that one of the reasons for the interest in porphyrin-like compounds in PDT resides in the Q-band absorption maxima that can be red-shifted as much as possible to fall into the so-called therapeutic window where the highest tissue permeability occurs. This is the reason that although all the most important spectral features of the whole spectrum for all the investigated compounds have been calculated, the attention has been mainly focused on the properties of the Q bands.

In order to select the most appropriate protocol for the analysis of the properties of the compounds under investigation, a preliminary benchmark of the maximum absorption wavelengths (λ_max_) of the two diagnostic porphyrin bands was carried out for both the **Fused-Fb** and **Fused-FbAu** compounds, as reported in Appendix A, with respect to the UV-vis experimental data [40,45]. Both compounds taken into account show an experimental UV-vis absorption spectrum characterized by a strong B band and a weaker Q band. Looking at the maximum wavelength of both bands, the best agreement with the experimental data was obtained using the M06 functional [46] that, however, gives a Q-band wavelength blue shifted with respect to the experimental value. The experimental spectra of the complexes named **Meso-Fb**, **Meso-Zn,** and **Meso-ZnAu** [41] were used to check if the M06 functional reproduces the **Meso** series well. The comparison between M06 and the experimental data can be found in Appendix A, which shows how the λ_max_ of both the B and Q bands is properly computationally reproduced. Then, the M06 xc functional was adopted to perform water solvent calculations for all the considered compounds. This choice, moreover, was in agreement with previous calculations carried out by some of us on porphyrin-like systems [13,47,48,49].

The main features of the computed two components of the Q band, generally denoted as Qx and Qy, are collected in Table 1 and Table 2 for all the examined **Fused** and **Meso** compounds, respectively. The plots of the corresponding Gouterman’s molecular orbitals are reported in Figure 2 for the species **Fused-Fb**, **Fused-FbAu**, **Fused-ZnAu**, and **Fused-PdAu**, whereas for the rest of compounds, their graphical representations are shown in Appendix A.

The analysis of the data reported in Table 1 and Table 2 shows that for the free-base **Fused-Fb** the low-intensity Qy component is H→L in nature (about 70%), while the Qx one involves mainly the H→L+1 (more than 60%) transition. For the free-base **Meso-Fb**, the same transitions give the main contribution, but their weight is lower. The coordination of the porphyrin to the Au(I) center through the formation of a C_NHC_-Au bond does not introduce any significant change to the position or to the intensity of the bands. When the Zn(II) and Pd(II) cations were included in the porphyrin cavity, only a small blue-shift of about few nanometers was observed, which was slightly more significant when Pd(II) was used as the inner metal ion. Finally, the contemporary presence of the peripheral Au(I) and the metal cation at the center of the cavity causes a few nanometers of displacement of the two Q-band components towards the blue region of the spectrum that, once again, is more substantial for the Pd complexes. Free-base porphyrin Q bands are, obviously, of a π → π* nature, with the MOs localized on the central porphyrin ring, as shown in Figure 2 and Appendix A, and it is evident that the gold atom as well as the carbene moiety are largely isolated from the aromatic system of the porphyrin core. The frontier MOs involved in the most important transitions are not influenced by the presence of both the peripheral Au(I) and central Zn(II) ions such as the optical properties of such compounds. An observable contribution of the metal center to the L and L+1 orbitals was noticed, instead, for the Pd(II) complexes.

The vertical excitation energies for the triplet states, together with the most significant molecular orbital contributions, are listed in Appendix A. The triplet states lying below the first singlet excited state are four. From the computed vertical transition energies for all the examined compounds, it clearly appears that the energy gap between the ground singlet and the lowest-lying triplet excited state (∆E_S0−T1_) is higher than the energy required to promote molecular oxygen in its ^1^∆_g_ state (0.98 eV), and consequently, they are potential candidates to act as photosensitizers by the type II mechanism. The generation of cytotoxic singlet oxygen by the type II mechanism relies on the possibility of the system to trigger the radiationless ISC process between a singlet excited state (S_n_) and a triplet state lying below it (T_m_).

As underlined above, the probability to populate the triplet states through an ISC depends on the spin–orbit coupling (SOC) values and the corresponding values of the singlet–triplet energy splitting ∆E_Sn−Tm_. Such energy gaps for all the examined compounds can be found in Appendix A. The presence of heavy atoms might influence the SOC values by the HAE that, eventually, boosts the probability of ISC. As there are two low-lying singlet excited states and there are four triplets below, all the ISC processes between the S_1-2_ and T_1-4_ states have been considered for all the compounds under investigation. All the calculated values are collected in Table 3. The two free-base porphyrins **Fused-Fb** and **Meso-Fb** present similar SOC values, with the largest matrix element being 0.4 cm^−1^ for the former and 0.3 cm^−1^ for the latter. The largest values correspond to the S_2_ → T_2_ and S_1_ → T_1_ transitions for **Fused-Fb** and **Meso-Fb**, respectively. These values are similar to that computed for Foscan^®^ (0.24 cm^−1^ for the S_1_ → T_1_ transition), a photosensitizer already used in PDT. The presence of the Zn(II) in the porphyrin core does not provoke any increase in the SOC values for both the **Fused** and **Meso** compounds. The computed values show that the HAE is quite irrelevant for the **Fused** porphyrin conjugated with Au(I), whereas a slight increase was observed for the corresponding **Meso** porphyrin.

This behavior is expected on the basis of the shapes of the frontier MO contour plots, as illustrated above, as well as that the largest SOC values are obtained for Pd(II) complexes. The largest values of the SOC elements for **Fused-Pd**, which are 17.4 and 17.1 cm^−1^, are due to the S_1_ → T_1_ and S_2_ → T_2_ transitions, respectively. The S_1_ → T_2_ and S_2_ → T_1_ transitions corresponds to the values of 18.5 and 17.1 cm^−1^, calculated for **Meso-Pd**. According to the El-Sayed rule, as a high ISC rate can be achieved if the non-radiative transition from the singlet to triplet excited states involves different molecular orbital configurations, a change in the contributing orbitals causes a change in the transitions originating the largest SOCs. The addition of a peripheral Au(I) complex to obtain the **Fused-PdAu** does not increase the SOC amplitudes but causes a change in the nature of the contributing orbitals, and the highest SOC values become 18.0 and 16.9 cm^−1^, corresponding to the S_1_ → T_2_ and S_2_ → T_1_ transitions. Compared to **Meso-Pd**, the amplitude of the SOCs for the corresponding Au(I) complex, **Meso-PdAu**, decreases, becoming 12.1 and 12.2 cm^−1^, and the involved transitions do not change. In summary, the presence of the external Au(I) complex, increases the SOC values only when conjugated with the **Meso** porphyrin but not enough to have a consistent enhancement of the singlet oxygen generation, whereas no synergistic effect on the SOC is detected when Au(I) is contemporarily present with Zn(II) and Pd(II). Thus, the obtained data show that peripheral Au(I) complexes do not improve the PDT efficiency from a photophysical point of view. However, as previously suggested, the peripheral Au(I) complexes can be beneficial to combine the dark cytotoxic activity of Au(I) with the photodynamic effect of the porphyrin when activated by light of a proper wavelength. Moreover, the Au(I) center can serve as a connecting metal ion to bind ligands that can improve other useful properties, such as water solubility or the active targeting of cancer cells.

## 3. Computational Details

All the calculations were carried out using Gaussian 16 software [50] adopting the density functional theory (DFT) approach. All the geometrical structures were optimized in implicit water solvent using the integral equation formalism variant of the polarizable continuum model (IEFPCM) [51,52] at the B3LYP-D3/6-31G** level of theory [53,54,55]. The excitation energies were obtained by performing linear response (LR) TDDFT calculations on the optimized structures of all the investigated compounds using the M06 functional [46] and 6-31G** basis sets for all the atoms except Au and Pd, while the effective core potentials SDD [56] were used together with the split valence basis set. This functional was chosen after a preliminary TDDFT benchmark in dichloromethane, the solvent used in the experiments, in order to select the best exchange and correlation functional that is able to reproduce the recorded absorption spectra [40,45] of **Fused-Fb** and **Fused-FbAu**. For this purpose, the structures of the compounds were optimized in CH_2_Cl_2_, and a few exchange and correlation functionals were taken into consideration, which were: B3LYP, PBE0 [57], CAM-B3LYP [58], M06, B3PW91 [59], and LC-wPBE [60]. The available experimental absorption spectra of **Meso-Fb**, **Meso-Zn**, and **Meso-ZnAu** were also used to check if the chosen protocol properly described the second class of porphyrins.

Subsequently, in order to mimic the cellular environment, UV spectra and SOC values were calculated in implicit water solvent. The spin–orbit matrix elements were evaluated using SOC-TD-DFT as implemented in the ORCA code [61,62]. A check was carried out to ascertain whether the outcomes of the two Orca and Gaussian codes used were superimposable when the energies of the coupled states were taken into consideration. Thus, relativistic corrections were computed by the zeroth order regular approximation (ZORA) at the ground-state-optimized geometries. Accordingly, the ZORA-DEF2-SVP and SARC-ZORA-SVP basis sets for the main and metal atoms, respectively, were employed, and the SOCs values were calculated as previously reported [13].

## 4. Conclusions

Porphyrins and their derivatives are, very likely, the most studied macrocycles for their use in photodynamic therapy (PDT) to treat different types of cancers because of their proper photophysical properties. A huge number of functionalizations have been tested, aiming at improving the properties of such compounds and enhancing the efficiency in generating singlet oxygen. Here, we have examined the photophysical properties of porphyrins containing peripheral N-heterocyclic carbene (NHC) Au(I) complexes. Both the presence of an external Au(I) ion bound to the NHC moiety and a Zn(II) or Pd(II) ion at the center of the tetrapyrrole ring were taken into consideration. Keeping in mind the underestimation of the absorption wavelengths by the TDDFT approach, the outcomes of our computational analysis show that all the compounds under examination, named **Fused** and **Meso** as a function of the way to link the NHC group to the porphyrin, are able to play their role as photosensitizers in type II PDT processes. Indeed, the maximum absorption wavelengths of the Q-band components fall in the so-called therapeutic window, allowing penetration into the tissue; the singlet–triplet energy gap is higher than that required to generate the cytotoxic singlet oxygen species; the calculated SOC elements for both the radiationless singlet–triplet couplings are similar or larger than that computed for the photosensitizer Foscan^®^ (0.24 cm^−1^ for the S_1_ → T_1_ transition), which is already used in PDT. The influence of the presence of the peripheral Au(I) center and of the Zn(II) and Pd(II) ions in the center of the porphyrin core in increasing the rate of the ISC attributable to the heavy atom effect, which causes a change in the SOC elements of two orders of magnitude only when the Pd(II) ion is involved. The presence of the external Au(I) ion, instead, causes an increase in the SOC values only when the NHC ligand is linked to the *meso* position of the porphyrin core (**Meso** series), but not enough to have a substantial enhancement of the singlet oxygen generation. No synergistic effect on the SOC is detected when Au(I) is contemporarily present with Zn(II) or Pd(II) ions. Plots of the four Gouterman’s orbitals confirm that the Q bands are of a π → π* nature, with the MOs localized on the central porphyrin ring and neither the NHC moiety and the Au(I) atom nor the Zn(II) center give a contribution to such orbitals. Only the Pd(II) ion participates in the LUMO and LUMO+1 orbitals, provoking the observed increase in the calculated SOC elements. The outcomes of the present computational analysis show that, even if it is well-established that the introduction of heavy atoms into a photosensitizer should have a direct influence on the rate of the ISC as a consequence of the HAE, no obvious relationship exists since both the identity and the position of the introduced heavy atoms determines the final effect.

## Data Availability

Not applicable.

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
