# Peer review of "Porphyrins and Metalloporphyrins Combined with N-Heterocyclic Carbene (NHC) Gold(I) Complexes for Photodynamic Therapy Application: What Is the Weight of the Heavy Atom Effect?"

_molecules, 2022, doi:10.3390/molecules27134046_

Round 1

Reviewer 1 Report

The Authors present a computational evaluation of the photophysical properties of two sets of porphyrins containing peripheral N-heterocyclic carbene Au(I) complexes and explain the influence of the presence of an external Au(I) ion bound to the NHC moiety and a Zn(II) or Pd(II) ion at the center of the tetrapyrrole ring.
The readability of the manuscript is good. Overall, the manuscript presents a very complete and consistent theoretical study. The method section provides a description of the experiments in good enough detail. In general, the conclusions are well supported by the discussion. The manuscript is suitable for publication once the authors address the following issues:

1.    For the theoretical values, the transition energy of the Soret band increases with metal substitution (from 411 to 406 nm), while for the experimental values, this relationship is reversed (Table S2). How can this be explained?

2.    Fig. S1. - please provide the RMSD value for the presented X-ray and DFT structures. Is it possible to present these structures for all of the studied compounds?

3.    In the manuscript, the Authors compare the results to Foscan (line 199, line 273). In my opinion, it may be inappropriate because Foscan is a chlorin derivative, not porphyrin. Thus, the results should also be compared with porphyrin-based, clinically-approved photosensitizer like Photofrin.

4.    The spin orbital constant is often used to describe spin-orbit coupling. However, it is also possible to calculate the fluorescence lifetime, which could be compared with experimental values from one of the cited publications (see rg. ref 28). If it is possible, please calculate the triplet lifetimes values in the revised version of the manuscript.

5.    I wonder how these results support the biological activity of selected compounds as a PDT agent. Did the Authors perform some experiments and can include eg. absorption spectra, triplet lifetimes, and ROS generation? These data may be crucial to prove the possible translation of this theoretical approach to the experimental screening of potential photosensitizers.

Reviewer 2 Report

The paper entitled „Porphyrins and Metalloporphyrins combined with N-heterocyclic Carbenes (NHC) Gold(I) Complexes for Photodynamic Therapy Application. What is the Weight of the Heavy Atom Effect?” by Stefano Scoditti, Francesco Chiodo, Gloria Mazzone, Sébastien Richeter and Emilia Sicilia reports the studies  of porphyrins by means of density functional theory and its time-dependent extension for their potential application in photodynamic therapy. Paper need major revision. All issues need to be considered are listed below:

1.      The selected literature requires significant modernization. For example, citations 2-4 are not representative of the topic and far too old. In the entire paper, almost 50% of citations are older than 10 years.

2.      Line 42 photo treatments (?)

3.      Line 123 no quotation. Classically, the range of 500-800 nm is considered as the therapeutic window, some authors suggest 850 as the upper limit - although in the opinion of the reviewer it is highly debatable.

The presented topic seems to be very interesting, but the research raises many questions, including:

A. Where did the selection of the presented structures come from? The literature analysis justifying the selection of these compounds has not been presented. If it has been guided by the authors' experiences in drug design or the relationship between structure and molecular target, it should be indicated. Moreover, there are many potential obstacles to the registration of PACT compounds with a heavy metal such as gold in the periphery. Above all, problems with high toxicity should be taken into account.

B. Based on the quoted literature, it is difficult to defend the relationship between the presence of a heavy atom and the solubility of the compound (line 73-77) in water. The topic of the water solubility of the analyzed compounds is presented in an unclear manner. In the opinion of the reviewer, the tested compounds would be insoluble in water. In such a case, it would be necessary to use carriers such as, for example, liposomes. They would affect the key parameters for PDT to an unpredictable degree.

C. In the case of medical applications, the relationship between the ability to generate ROS and therapeutic efficacy is not so obvious. Compounds with a low ability to generate ROS have often turned out to be extremely effective in antimicrobial therapy. This is due to many factors that are ignored in this work.

D. The presented data in the paper is difficult to validate. The reviewer understands that the appropriate calculations have been carried out with the use of the indicated program, but what was the method of controlling the results?

E. The question contained in the title of the article is confusing. The answer to them has been known for many years, and on the other hand, even in the conclusions there is no direct reference to it. Such a question would be more justified in a review or an extensive research article in which the influence of a heavy atom would be compared, e.g. for various subgroups of porphyrinoids substituted analogously in the periphery.

Reviewer 3 Report

This paper systematically investigated the excitation state properties of N-heterocyclic carbenes (NHC) metal complexes. The non-relativistic excitations are described by TDDFT. The size of the basis set is around the minimum acceptable level. It is significant that the relativistic effect was investigated by the zeroth-order regular approximation (ZORA), which is the two-component relativistic method. Lots of studies just included scalar relativistic effect or even just RECP correction. Therefore, this paper will provide more accurate results on the relativistic effect. There are a few questions about the computational details. This paper can be published if the following questions are addressed properly.

1.     line 148 and line 153, ΔE should be excitation energies.

2.     Symbols of physical properties, such as energy E and wavelength λ, should be italic.

3.     Line 149 and 154, there should be no comma between “oscillator strengths” and “f”.

4.     Implicit solvation model was used. It would be helpful to mention the exact model in the Computational Details section. Is it IEFPCM or SMD?

5.     Does the excited-state calculation use linear response (LR) solvation? Or is it a state-specified (SS) solvation in which external iteration is done? It would be helpful to mention this in the Computational Details section.

6.     The author claimed a Type-II mechanism for generating singlet O2. The triplet states should be higher than 0.98 eV. Meanwhile, high ISC rate between S and T states. What is the criterion for a high ISC rate? Could the author give some examples of typical compounds with high ISC rates in the discussion or conclusion section?

Author Response

please, see the attachment

Round 2

Reviewer 1 Report

Authors replied properly to my suggestions

Reviewer 2 Report

Paper can be published